# Mitigating carbon footprint for knowledge distillation based deep learning model compression

**Kazi Rafat[1], Sadia Islam[1], Abdullah Al Mahfug[1], Md. Ismail Hossain[1], Fuad Rahman[2], Sifat Momen[1], Shafin Rahman[1]\*, Nabeel Mohammed[1]**

**1** Apurba NSU R&D Lab, Department of Electrical and Computer Engineering North South University, Dhaka, Bangladesh, **2** Apurba Technologies, Dhaka, Bangladesh

\* shafin.rahman@northsouth.edu

**Data Availability Statement:** All data and computational codes are released in the public GitHub repository (https://github.com/King-Rafat/STKD_CFMitigation).

## Abstract

Deep learning techniques have recently demonstrated remarkable success in numerous domains. Typically, the success of these deep learning models is measured in terms of performance metrics such as accuracy and mean average precision (mAP). Generally, a model's high performance is highly valued, but it frequently comes at the expense of substantial energy costs and carbon footprint emissions during the model building step. Massive emission of $CO_2$ has a deleterious impact on life on earth in general and is a serious ethical concern that is largely ignored in deep learning research. In this article, we mainly focus on environmental costs and the means of mitigating carbon footprints in deep learning models, with a particular focus on models created using knowledge distillation (KD). Deep learning models typically contain a large number of parameters, resulting in a 'heavy' model. A heavy model scores high on performance metrics but is incompatible with mobile and edge computing devices. Model compression techniques such as knowledge distillation enable the creation of lightweight, deployable models for these low-resource devices. KD generates lighter models and typically performs with slightly less accuracy than the heavier teacher model (model accuracy by the teacher model on CIFAR 10, CIFAR 100, and TinyImageNet is 95.04%, 76.03%, and 63.39%; model accuracy by KD is 91.78%, 69.7%, and 60.49%). Although the distillation process makes models deployable on low-resource devices, they were found to consume an exorbitant amount of energy and have a substantial carbon footprint (15.8, 17.9, and 13.5 times more carbon compared to the corresponding teacher model). The enormous environmental cost is primarily attributable to the tuning of the hyperparameter, Temperature ($\tau$). In this article, we propose measuring the environmental costs of deep learning work (in terms of GFLOPS in millions, energy consumption in kWh, and $CO_2$ equivalent in grams). In order to create lightweight models with low environmental costs, we propose a straightforward yet effective method for selecting a hyperparameter ($\tau$) using a stochastic approach for each training batch fed into the models. We applied knowledge distillation (including its data-free variant) to problems involving image classification and object detection. To evaluate the robustness of our method, we ran experiments on various datasets (CIFAR 10, CIFAR 100, Tiny ImageNet, and PASCAL VOC) and models (ResNet18, MobileNetV2, Wrn-40-2). Our novel approach reduces the environmental costs

**Funding:** The author(s) received no specific funding for this work.

by a large margin by eliminating the requirement of expensive hyperparameter tuning without sacrificing performance. Empirical results on the CIFAR 10 dataset show that the stochastic technique achieves an accuracy of 91.67%, whereas tuning achieves an accuracy of 91.78%—however, the stochastic approach reduces the energy consumption and $CO_2$ equivalent each by a factor of 19. Similar results have been obtained with CIFAR 100 and TinyImageNet dataset. This pattern is also observed in object detection classification on the PASCAL VOC dataset, where the tuning technique performs similarly to the stochastic technique, with a difference of 0.03% mAP favoring the stochastic technique while reducing the energy consumptions and $CO_2$ emission each by a factor of 18.5.

## Introduction

Deep learning methods are making significant advancements in solving problems deemed infeasible after years of tremendous effort by the artificial intelligence (AI) community [1]. The success obtained by the deep learning techniques has come in different substantial areas of AI such as object detection [2–5], image classification [6–8], natural language processing (NLP) [9, 10], medical imaging and processing [11–13] etc. Typically, models are fine-tuned to increase performance by a small margin to achieve state-of-the-art results. However, the efficacy of deep learning models endure substantial energy costs and produces a considerable carbon footprint that can contribute to climate change that is harmful to life on this earth. We know that Carbon dioxide ($CO_2$), a greenhouse gas, is a major contributor to global warming [14] that has a direct detrimental effect on agriculture [15], forestry [16, 17], species loss [18], sea-level rise [19], electricity demand [20], human amenity, human morbidity, natural disasters [21], construction, water supply, urban infrastructure, infectious diseases [22] and so on. Nevertheless, according to Strubell et al. [23], a basic NLP pipeline with tuning produces over twice the amount of $CO_2$ produced by the average American in one year. Being in oblivion, deep learning researchers, focus more on creating better models and prioritize model performance over the society and the environment. In this paper, we investigate the carbon footprint associated with a deep learning strategy called knowledge distillation and suggest a plausible solution to mitigate this problem.

Knowledge distillation is a well-known tool for model compression. Deep learning models are generally large architectures containing millions of parameters (weights). Model compression aims to design a relatively lighter version of the original large version without compromising the overall performance [24]. From the literature, larger and lighter models are called the teacher and student models, respectively [25]. Following the knowledge distillation process, a teacher model shares its knowledge with student models so that the students not only learn from the dataset but also directly from the teacher. In many real-life applications, especially in mobile or edge devices with computational and hardware constraints, deploying the lighter student model (instead of the larger teacher model) can be a pragmatic solution. At present, Hinton's knowledge distillation [26] can be considered the most widely used model compression technique due to its efficacy and simplicity. The distillation process softens output logits of the teacher model to transfer information from the teacher to the student model. The student model uses the logit distribution of the teacher model across all training instances as an altered training set. Soft outputs of the teacher model act as the ground truth (supervision) while learning the student model. This auxiliary information provided by the teacher (contained in the soft outputs) is known as dark knowledge [26, 27]. Dark knowledge also contains

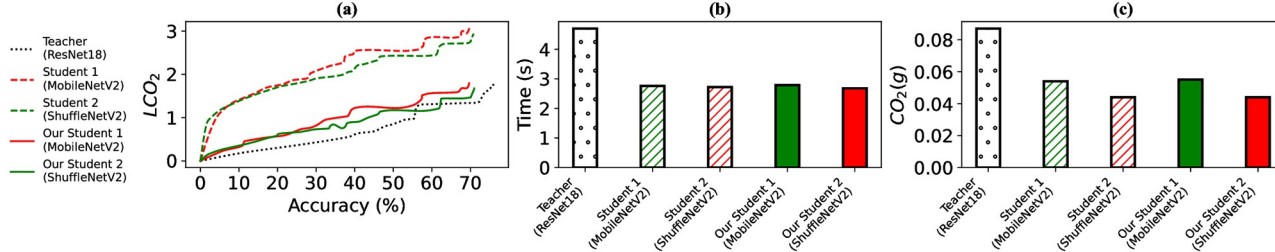

**Fig 1.** Illustration of carbon footprints used by different deep models while **(a)** training on CIFAR 100 (in log scale) and **(b-c)** inferring on evaluation set. ResNet18 is a deeper model with 11.2M parameters, resulting in higher inference time (4.7 sec.) and $CO_2$ emission (0.087 g). To minimize this, using ResNet18 as a teacher, we train two student models, MobileNetV2 (student 1) and ShuffleNetV2 (student 2), following the traditional KD process. This training costs significant carbon footprints (red and green dashed curves in **(a)**) with an accuracy increment from learning the teacher model (black dotted curve in **(a)**). However, as expected, both students consume less time and $CO_2$ during inference (red and green shaded bars in **(b)** and **(c)**). We aim to reduce the training cost and $CO_2$ production of the KD process while using the same students (red and green solid curves in **(a)**) and maintain similar accuracy and inference costs (solid red and green bars in **(b)** and **(c)**) in comparison with the costly KD training.

helpful information of class similarity in a dataset that is helpful for the model to learn [26]. For proper knowledge transfer from the teacher to the student, the softening process requires tuning a hyperparameter named Temperature, $\tau$. Hinton et al. [26] used a $\tau$ value between 2 and 20 inclusively. However, $\tau$ can still be increased to test with softer outputs. An improper value of $\tau$ produces inefficient models with a lackluster performance. Therefore, to attain a suitable student model, knowledge distillation frameworks must iterate several times with different values of the hyperparameter $\tau$. Preparations and tuning around such frameworks raise substantial financial and environmental costs by engaging longer training time and higher carbon footprint. In this paper, we estimate the overhead of existing methods and propose a mitigation process by using a stochastic process related to a hyperparameter instead of a grid search [28] (see Fig 1).

To tackle these problems, this paper proposes a stochastic technique of assigning the hyperparameter $\tau$. This process of assigning a stochastic temperature facilitates training while drastically reducing costs and preserving the performance of the assembled student model. This system of training a model compression system can achieve comparable performance to a fine-tuned system while reducing the time, power costs, and carbon footprint. In summary, our contributions are as follows:

- We investigate the environmental costs (carbon footprints) for deep learning model compression using knowledge distillation. It establishes that the cost is not insignificant, and researchers should be aware of this matter considering the ongoing climate change movement.

- We propose a stochastic approach to mitigate the carbon footprint issues related to the knowledge distillation process. It helps to produce satisfactory performance (compressed model without compromising accuracy), reducing energy consumption. One can easily integrate the method with any knowledge distillation pipeline without high overhead costs.

- We perform extensive experiments on different student-teacher model combinations based on ResNet18 [29], MobileNetV2 [30], and ShuffleNetV2 [31] architectures. We report results on both object recognition and detection problems using CIFAR10 [7], CIFAR 100 [7], Tiny Imagenet [32] and PASCAL VOC [33, 34] datasets. For all experiments, we compare carbon footprints (GFLOPs count in millions, energy consumption in kWh, and $CO_2$ equivalent in grams) between existing and our proposed methods.

## Related works

### Energy cost for deep learning

Recent technological advancements in hardware, model development, and training frameworks have stimulated the rise of highly accurate deep learning models [9, 35–37]. Such models, however, are expensive in terms of computational resources and energy consumption, resulting in a massive increase in their carbon footprint. Recent studies have investigated energy usage and the environmental impact of deep learning models to raise awareness regarding this situation. Strubell et al. [23] deduced that training large NLP frameworks trends towards a hefty carbon footprint that is detrimental to the environment. Another study [38] discovered that hyper-parameter tuning of large frameworks has a significant carbon footprint, which has a negative impact on climate change. However, such factors are rarely considered because most papers published in major venues aim for accuracy rather than computational efficiency [39]. Parcollet and Ravanelli [40] estimated a huge carbon footprint related to training speech recognizers. Tamburrini [41] discussed the ethical implications for AI researchers, manufacturers, and industries. Given the significance of this problem, very few studies have been aimed at mitigating this issue. Schwartz et al. [39] identified the primary reasons contributing to a high carbon footprint and also proposed the efficiency metrics in deep learning research for proper energy utilization. Anthony et al. [42] also tackled this problem and developed a tool to track and estimate the carbon footprint produced by deep learning frameworks. This study aims to assess student-teacher frameworks' carbon footprint for model compression.

### Knowledge distillation methods for model compression

Buciluă et al. [43] first introduced the term 'model compression' to compress large complex ensembles into smaller and faster models without significant performance loss. Knowledge distillation (KD) is a popular strategy to address the model compression problem. KD trains a smaller network with the help of larger models and ground labels. Lei and Caruana [44] demonstrated that shallow feed-forward nets could learn the complex functions previously learned by deep nets and achieve performances previously only achievable with deep models. Later, Hinton et al. proposed a KD-based model compression method [26] that distills a teacher's knowledge using a special loss by softening the outputs of the teacher to a student model, which is widely used for compression tasks. Further improvements in this line of investigation followed Hinton's distillation process. Liu et al. [45] improved novelty in Hinton's distillation approach by distilling pixel-wise and structure distillation schemes. Park et al. [46] have resorted to a structured comparison of the teacher and student to improve Hinton's distillation. In this paper, we investigate knowledge distillation in object classification and detection in the context of model compression.

### Applications of KD

Recently, knowledge distillation has been successfully applied to solve complex machine-learning problems, mainly image classification. Li and Hoem [47] used knowledge distillation to make a unified vision system that utilizes the knowledge in models to add new capabilities to the system without requiring training data. Liu et al. [48] focused knowledge distillation on Multi-Label Classification, which focuses on creating a Weakly Supervised Model (WSM) to distill its knowledge for better classification. Wang and colleagues [49] put to use knowledge distillation to explain and interpret image classifiers by developing smaller image classifiers that are easily explainable. Peng et al. [50] utilizes correlation between

images to distill knowledge for improvement in image classification. Knowledge distillation and model compression are currently being used for object detection, as models tend to be large and require a costly runtime. Chen et al. [51] employed knowledge distillation to make efficient object detection models with a similar performance by innovating model compression to suit object detection. Li and colleagues [52] distilled models for improving a photon-limited object detection framework. Guo et al. [53] showcases a method of online knowledge distillation where all DNNs are considered as students for a more effective distillation for object detectors. Dai and his [54] team used an alternative form of knowledge distillation for object detection that uses three knowledge distillations for proper knowledge transfer. The routine unavailability of training data due to privacy issues has resulted in a recent increase in the popularity of data-free knowledge distillation, where models are trained through distillation without the training data. Chawla et al. [55] utilizes knowledge distillation for data-free object detection such that the generated data enables better distillation of knowledge. Zhang and colleagues [56] developed a generator network for producing artificial training data for image super-resolution using progressive knowledge distillation. In their work, Yin and team [57] discussed their data production technique, Deep Inversion, which produces training data from a trained network by inverting the model without prior knowledge of the data. Choi et al. [58] utilizes meta-data from the teacher network to generate artificial data using a generator that was not trained on training data for knowledge transfer to a smaller network. Note that knowledge distillation is also readily used in n-shot learning techniques. Nayak et al. [59] proposed techniques that allow zero-shot knowledge distillation without data samples or meta-data. Michaeli and Storkey [60] also proposed a novel method to transfer knowledge to students using a generative adversarial network for a zero-shot knowledge transfer. In this paper, we propose a stochastic approach of assigning temperature to knowledge distillation frameworks to eliminate the hefty requirement of the tuning hyper-parameters temperature.

## Methodology

Recently, various model compression techniques have been proposed for deep learning, primarily utilizing information from the teachers to train a smaller student better (knowledge distillation) [26, 43, 44], modifying networks to reduce the model size (pruning) [61–63] or trading precision for a smaller size (quantization) [64, 65]. In this section, we define the problem, the drawbacks of the current solution, a proposal for improvements, and relate carbon footprint issues with AI ethics.

### Problem formulation

Suppose, $\mathcal{F}_m(X; W_m)$ is a deep neural network model where $X$ and $W_m$ represent the input image and model parameters/weights, respectfully. Let us consider, number of parameters, $|W_m|$ is relatively high, meaning a deeper model which obviously performs well in a recognition or detection tasks. However, computation cost during inference is high because of its larger size. To reduce the computational cost, our aim is to design a compressed model, $\mathcal{F}_s(X; W_s)$ where, number of parameters $|W_s|$ is significantly smaller than $|W_m|$, i.e., $|W_s| << |W_m|$ but performance (accuracy or mAP) of both $\mathcal{F}_m(:; W_m)$ and $\mathcal{F}_s(:; W_s)$ models are similar. The architecture of $\mathcal{F}_s(:; W_s)$ and $\mathcal{F}_m(:; W_m)$ can be same or different. To train $\mathcal{F}_s(:; W_s)$, we can use logits of the model $\mathcal{F}_m(:; W_m)$ for knowledge transfer. Therefore, in this setup, $\mathcal{F}_m(:; W_m)$ and $\mathcal{F}_s(:; W_s)$ are called teacher and student models, respectively.

## Solution using Knowledge distillation (KD) [26]

KD is a well-known tool to compress a possibly larger (teacher) model, $\mathcal{F}_m(X; W_m)$ to a smaller (student) model $\mathcal{F}_s(X; W_s)$. Here, the input space $X$ belongs to label space $y \in \mathbb{R}^n$, where $n$ represents input the number of classes available in the dataset. We assume a supervised classification setting where model outputs are $a_m = \mathcal{F}_m(X; W_m)$, $a_s = \mathcal{F}_s(X; W_s)$; $\{a_m, a_s \in \mathbb{R}^n\}$ before activation such that $a_m$ and $a_s$ represent the un-activated output of teacher and student models, respectively. The student model outputs are activated using the softmax function [66] to produce a hard embedding $h_s$ such that $h_s(i) >> h_s(j)$, $\{1 <= i, j <= n, i \neq j\}$ where the logits are probability distributions $\sum_{i=1}^{n} h_s(i) = 1, h_s(i) \in \mathbb{R}^+$. However, transfer of knowledge is done through proper softening of the output of the teacher $a_m$ and student model $a_s$, which is done by a modified softmax activation containing a hyper-parameter $\tau$:

$o_z = \sigma(a_z) = \frac{e^{a_z(i)/\tau}}{\sum_{j=1}^{n} e^{a_z(i)/\tau}}$ where, $a_z(i), a_z(j) \in a_z \subset \mathbb{R}$, $|a_z| = n$, $1 <= i, j <= n$. Here the variable $\tau$ is a hyper-parameter crucial for softening of logits $o_s$, which theoretically ranges from $1 < \tau < \infty$. The activation provides softened outputs (probability distribution) for $o_s$ (student) and $o_m$ (teacher) using $z = s$ or $z = m$. Here, $o_s = [o_s(1), o_s(2), o_s(3), \ldots, o_s(n)]$, where, $\sum_{i=1}^{n} o_s(i) = 1$, $o_{sn} \in \mathbb{R}^+$ and $o_m = [o_m(1), o_m(2), o_m(3), \ldots, o_m(n)]$, where, $\sum_{i=1}^{n} o_m(i) = 1$, $o_m(n) \in \mathbb{R}^+$.

The acquisition of soft labels helps utilize the dark knowledge stored in the teacher model for proper knowledge transfer to the student model. The outputs $h_s, o_s, o_m$ and label $y$ are used to provide gradient information for the student to learn better using the distillation loss which is back-propagated to update model parameters. The distillation loss ($\mathcal{L}_{DL}$) consists of the weighted union of the softmax Cross Entropy ($\mathcal{L}_{CE}$) [67] and the Kullback-Liebler Divergence ($\mathcal{L}_{KLD}$) [68] loss with an added weight.

$$\mathcal{L}_{CE}(y, h_s) = -\sum_{c=1}^{n} y(c) \log h_s(c), \qquad \mathcal{L}_{KLD}(o_s, o_m) = \tau^2 \sum_{c=1}^{n} o_m(c) \log \frac{o_s(c)}{o_m(c)} \qquad (1)$$

Knowledge distillation varies from a traditional classification because of $\mathcal{L}_{KLD}$, which facilitates the student to learn. The hyper-parameter $\tau$ usually gets a value within the range $\tau$ = 2 to 20. $\tau$ contributes to softmax distributions with different levels of softness. Proper softening of the logits enables better distribution of information to the student, $S$. As a result, the hyperparameter temperature is essential in knowledge distillation for proper knowledge transfer. For instance, a larger model might require more softening, i.e., a higher value of $\tau$ is expected. Additionally, knowledge distillation requires another hyper-parameter $\alpha$ which is used to assign weights to the Cross-Entropy and Kullback-Liebler loss functions for proper parameter tweaking by gradient descent. Fine-tuning is required in some cases for perfect model distillation. Finally, the distillation loss is formulated as follows:

$$\mathcal{L} = \underbrace{\alpha \, \mathcal{L}_{KLD}(o_s, o_t)}_{\tau=2 \text{ to } \infty} + \underbrace{(1 - \alpha) \, \mathcal{L}_{CE}(y, h_s)}_{\tau=1} \qquad (2)$$

The process of knowledge distillation is briefly shown schematically in Fig 2.

## Issues with knowledge distillation

The Kullback-Liebler loss provides additional gradient information to the distillation loss that transfers a teacher's knowledge to a student. Properly softening the hyper-parameter Temperature, $\tau$, is required to produce the softening effect that introduces dark knowledge for the

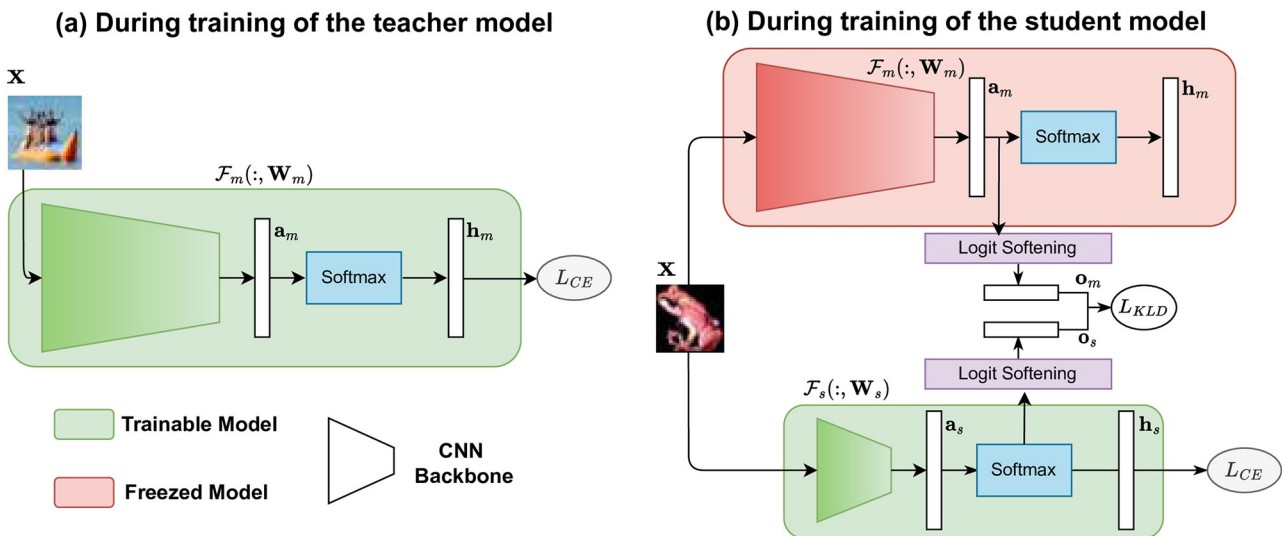

**Fig 2. Block diagrams of KD architectures while training teacher, $\mathcal{F}_m(:, W_m)$ and student, $\mathcal{F}_s(:, W_s)$ models. (a)** Given input $X$, trainable $\mathcal{F}_m(:, W_m)$ (indicated as green) learns to produce logits $h_m$ after a softmax activation. Cross-Entropy loss ($\mathcal{L}_{CE}$) is used to train the teacher model. **(b)** Trainable student model, $\mathcal{F}_s(:, W_s)$ (indicated as green) leanrs from a frozen (indicated as red) teacher, $\mathcal{F}_m(:, W_m)$. The teacher and student produce unactivated logits $a_m$ and $a_s$, respectively. $a_s$ are activated using a softmax containing the hyperparameter $\tau$ producing soft logits $o_m$. Similarly, $o_s$ is produced for the teacher. The soft logits are used to calculate KLD loss ($\mathcal{L}_{KLD}$). The hard logits from the student $h_s$ are further used to calculate $\mathcal{L}_{CE}$. $\mathcal{L}_{KLD}$ acts as additional supervision for the student to learn better.

student to consume. An unsuitable value of $\tau$ might harm students' training. In contrast, Fig 3(a) shows that adjusting/tuning a temperature for knowledge distillation may lead to performance enhancement. Therefore, hyper-parameter $\tau$ requires fine-tuning through grid search [69]. Fine-tuning such a hyper-parameter requires additional time and bears huge computation and energy costs. To facilitate this tuning, KD training may produce a huge carbon footprint, especially for a vast dataset. In this way, model tuning contributes to climate change and becomes a concern for ethical AI.

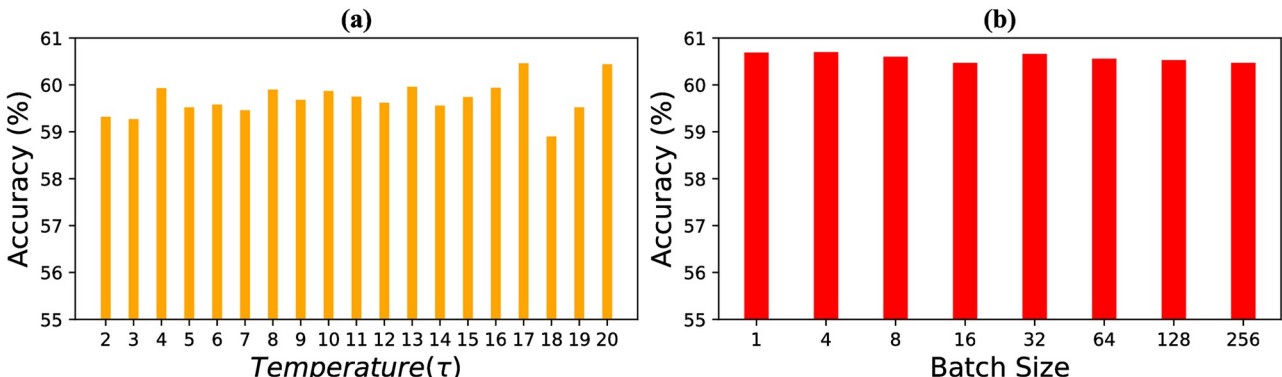

**Fig 3.** Impact of using different $\tau$ for ResNet18 (teacher) and MobileNetV2 (student) models using **(a)** Tiny ImageNet datasets. We notice a significant performance variance across different $\tau$ values. **(b)** The impact of using different batch sizes for our proposed stochastic solution. Similar performance across different batch sizes shows that our proposal does not depend on training batch sizes.

## Carbon footprint for knowledge distillation

In this subsection, we discuss the carbon footprint estimation process used in this paper and analysis the energy consumption of the traditional KD process.

**Measurement criteria.** We use three criteria [42] to measure the carbon footprints of the KD process: FLOPs count in millions, energy consumption in kWh, and $CO_2$ equivalent in grams. Now, we briefly summarize those criteria below.

*FLOPs count*. FLOPs (Floating-point Operations) estimate the number of additions and multiplications a model performs for a singular instance that the model receives. FLOPs count for a model is a vital efficiency measure since it showcases the number of calculations needed for a prediction. It also determines the sophistication of the model and the time it consumes. Large models tend to make more calculations, i.e., higher FLOPs count.

*Energy*. Energy consumption directly correlates an experiment with the amount of carbon footprint it produces. The power collected in the study is the total power of the computer components, mainly CPU, GPU, and DRAM. The total power consists of the overall power used by the computer components and not the power consumed by a deep learning experiment. As a result, energy/power consumption may vary for a similar experiment. Accurately attaining dynamic power or power for particular work is challenging and, at times, infeasible. Due to limited components tracked for power, the consumed is multiplied by the Power Usage Effectiveness (PUE). The PUE is the ratio of the total power used by the data center to the energy used to make a computation. We use PUE = 1.67, the global average for the year 2019 [70], because of the infeasibility of calculating the PUE in the locality where experiments are conducted.

First, the instantaneous power, $p_i$, or the power consumed per second, is measured in kWh for a duration of one training epoch. We denote $t_e$ as total number of epochs. Using the instantaneous power, the average power $p_{avg}$, can be calculated as follows:

$$p_{avg}(kWh) = \frac{1}{t_e} \sum_{i=1}^{t_e} p_i \qquad (3)$$

Next, the total power consumed by the computer components for one singular epoch $e$ on device $d$ is summed over the total number of training epochs $E$. The consumption for all the computational devices $D$, using the average power $p_{avg}$, requiring $t_e$ time for one training epoch, and multiplied with the PUE shown in Eq. below:

$$p_t(kWh) = PUE \times \sum_{e \in E} \sum_{d \in D} p_{avg} \times t_e \qquad (4)$$

*$CO_2$ equivalent*. To estimate the carbon emission, we calculate $CO_2$ equivalent ($c_t$) by the product of the carbon intensity $c_i$ of the area where the experiment is being carried out, and the energy consumed $p_t$, of that experiment.

$$c_t(g) = p_t \times c_i \qquad (5)$$

**Energy estimation.** In Table 1, we report the carbon footprints of the KD process. Large learning models (like the teacher) have a better learning capacity (Acc. 95%) due to many trainable parameters. However, such models usually have a large model size and take longer time for prediction, producing a vast carbon footprint. In contrast, lightweight models (like the student) are relatively small in architecture and faster during inference. However, such models lack performance (Acc. 90.6%) compared to large models. Model compression techniques such as KD are used to train lightweight models to mimic large models' performance.

**Table 1. Carbon footprints of Teacher (ResNet18) and Student (MobileNetV2) model before KD (top two rows), traditional KD [26] and `Ours` approach on three different datasets.** ↑ (↓) means higher (lower) is better.

| Method | CIFAR 10 | | | CIFAR 10 | | | Tiny ImageNet | | |
|---|---|---|---|---|---|---|---|---|---|
| | Accuracy (%) ↑ | Energy (kWh) ↓ | $CO_2$_eq (g) ↓ | Accuracy (%) ↑ | Energy (kWh) ↓ | $CO_2$_eq (g) ↓ | Accuracy (%) ↑ | Energy (kWh) ↓ | $CO_2$_eq (g) ↓ |
| Teacher | **95.04** | 0.28 | 74.25 | **76.03** | 0.25 | 65.55 | **63.39** | 1.43 | 413.60 |
| Student | 90.52 | **0.16** | **45.02** | 67.80 | **0.16** | **44.23** | 59.64 | **0.68** | **182.53** |
| KD [26] | **91.78** | 3.99 | 1173.44 | **69.70** | 3.80 | 1170.40 | 60.46 | 19.09 | 5595.31 |
| Ours | 91.67 | **0.21** | **61.76** | 69.40 | **0.20** | **61.60** | **60.53** | **1.02** | **304.88** |

A student model taught using a good teacher model performs surprisingly better when fine-tuned (Acc. 91.8%). However, the KD technique has disadvantages because it consumes about 25 times more energy and produces substantial carbon footprints than training a similar base model (see Table 1). For instance, image classification with KD on a standard Tiny ImageNet dataset using a ResNet18 teacher and a ShuffleNetV2 student produces 5595.31g of $CO_2$ (see Table 4). It consumes 19.09 kWh of energy for a fully tuned experiment, equivalent to a car traveling 46.5km. Even smaller datasets regularly used as benchmarks, such as CIFAR 10, produce a considerable carbon footprint, about 1170.21g, and consume about 3.99 kWh of power (see Table 4) for a tuned experiment using a prominent teacher, small student combination. Modern usage of knowledge distillation, such as in data-free knowledge distillation, also produces a hefty amount of $CO_2$ in both data production and model compression. Data-free knowledge distillation using the Contrastive Model Inversion technique produces 1353.7g of $CO_2$ (see Table 6) on the small and standard CIFAR 10 dataset. It yields a nearly similar amount of $CO_2$ (1369.46g) (see Table 6) when using Deep Inversion as the data-generating technique. Recently, knowledge distillation is also prevalent in object detection settings. Single-Shot Multibox Detection (SSD) with a VGG16 teacher and MobileNetV2 student consumes 1.69 kWh energy for a perfectly tuned model. It produces 9235.39g of $CO_2$ (see Table 4), estimated to be equal to a car traveling for 78.66km.

**Stochastic solution.** As discussed earlier, the KD process is costly because of the hyperparameter $\tau$ selection that ranges theoretically $2 <= \tau <= \infty$. We now discuss different strategies to select $\tau$ efficiently and propose our recommendation.

*Grid vs. random search.* Grid search is a hyperparameter tuning process that iterates over every combination of pre-defined hyperparameter values. Its search space is relatively high, which can contribute to carbon footprints significantly. In contrast, random search randomly selects a combination of hyperparameters from a search space. Nonetheless, random search has shown better performances in some cases where increasing the search space finds a better combination when the important hyperparameter is unknown [28]. However, hyperparameter $\tau$ controls the level of softening during KD, which is sensitive to performance. As a result, random search is unsuitable in this scenario. $\tau$ is usually tuned through costly grid search.

*Dynamic temperature, $\tau$.* Several works attempt to find dynamic $\tau$ for better performance. Jafari et al. [71] use a dynamic temperature, but the training procedure consists of two stages which are computation heavy. It also requires choosing a max temperature at the beginning of the training that can be treated as another hyperparameter. Liu et al. [72] employ dynamic hyperparameters by using another model/network to update parameters according to validation performance which helps to improve accuracy. Li et al. [73] also use a two-layer MLP for the same reason. All these techniques focus on maximizing performance. With the cost of considerable energy for training, they achieve a minuscule rise in performance. We argue that such a process is not green computation friendly.

*Our proposal.* Deep networks are trained using random batches, which produce hard output vectors ($h_m$ for teacher or $h_s$ for student). Hyperparameter, $\tau$ controls the softening of each element inside $h_m$ or $h_s$. Knowledge distillation utilizes dark knowledge for information transfer which is the information in the soft outputs. The logit information is blurred or smoothed out if the temperature is too high. If the temperature is too low, the embedding becomes sparse [74]. As a result, stronger models usually need higher temperatures since they produce sparse logit embeddings (high probability for target class) and vice versa. For better knowledge transfer, this softening should vary from instance to instance. Considering the random instances inside a batch and different batches should produce random levels of embedding sparsity, different $\tau$ is expected to match instances' diversity. A constant temperature throughout the training cannot model such randomness during the learning process. Grid/random search considers a single $\tau$ for the entire training. Similarly, dynamic $\tau$ use the same value from many consecutive epochs. For this reason, they must repeat the train-validation loop employing multiple $\tau$, which increases the training cost. Therefore, we recommend using a random $\tau$ for each batch within a range (say [2, 20]). It will also exclude the necessity of tuning $\tau$. The random $\tau$ works with random batches since every random batch requires random softening. In Fig 3(b), we show the impact of using different batch sizes considering random $\tau$ within [2, 20]. We notice that our stochastic proposal works consistently across batch sizes. It proves the necessity of using random $\tau$ during training. Evidently, through empirical evidence, we establish that the stochastic temperature performs similarly or, at times, better in the case of traditional knowledge distillation in comparison to tuning. Along with a similar performance, the stochastic technique is efficient as it produces significantly less carbon footprint and consumes less energy than classical grid-search or random-search (see Table 1).

Considering a sample input with its label, $(X, y)$ which is an element in dataset $D$, a student model $\mathcal{F}_s(X, W_s)$ produces soft logits $o_s$ from unactivated logits $a_s$. The training objective is to match the soft logits $o_s$ to the soft logits $o_m$, of a large teacher $\mathcal{F}_m(X, W_m)$, from the teacher's own unactivated logits $a_m$, softened by a temperature $\tau$, using an altered softmax $o_z = \sigma(a_z) = \frac{e^{a_z(i)/\tau}}{\sum_{j=1}^{n} e^{a_z(j)/\tau}}$ where, $a_z(i), a_z(j) \in a_z \subset \mathbb{R}, \ |a_z| = n, \ 1 <= i, j <= n.$ Knowledge distillation uses a KL divergence loss as additional information from the teacher given by $\mathcal{L}_{KLD}(o_s, o_m) = \tau^2 \sum_{c=1}^{n} o_m(c) \log \frac{o_s(c)}{o_m(c)}$ (as shown in Eq 1) softened and weighted by $\tau$. Considering the model is learning for $\omega$ epochs and with a constant $\tau$ as in traditional KD, the teacher output $o_m$, remains the same but the student model output $o_s$ changes to optimize towards $\mathcal{L}_{KLD}(o_s, o_m)[\omega_p] < \mathcal{L}_{KLD}(o_s, o_m)[\omega_q],$ such that $\omega_p > \omega_q$. In this setting, for the input $X$ the target $o_m$ is constant throughout the training cycle of the student due to a fixed $\tau$. In our proposed stochastic approach, the value of $\tau$ is varied stochastically for every batch, thus ensuring that the target $o_m$ for an input sample $X$ remains varies. This variety in $o_m$ may prevent $\mathcal{F}_s(X, W_s)$ from learning to optimize from a single $o_m$ and therefore likely providing a regularizing effect. This may eliminate the need to search for a fixed validated $\tau$.

**Green computing and ethical AI.** Along with its benefits, AI's usage and development pose severe ethical concerns regarding multiple individual/societal and environmental issues. Deep learning often requires the distribution of sophisticated hardware that requires expensive materials to develop that are running out [75]. Moreover, AI and its development require a lot of computation-hungry methods, which consume substantial computational power and lead to the depletion of natural resources and valuable materials. Furthermore, energy consumption is very high and increasing steadily in the case of computer vision and NLP [23, 39]. Additionally, the power consumption for such AI methods adds up to a large global carbon footprint that is highly harmful to the environment and contributes heavily to climate change.

AI development is increasingly unsustainable due to the significant usage of these natural resources. To tackle these problems, developers and researchers should have equitable access to resources, figure out or use techniques concerned with Green AI, and report the cost for training and tuning deep learning models [23]. This paper may assist in developing strategies that deal with making the hyper-parameter tuning in knowledge distillation and adding to Green AI. Here are some recommendations concerning knowledge distillation and green computing: *(1)* KD method-related papers should report carbon footprint metrics (FLOPs count, energy consumption, and $CO_2$ equivalent) regarding model training. *(2)* In addition to performance (accuracy/mAP score), the research community should establish some green index to start a healthier competition and promote energy-efficient methods.

## Experiment

### Setup

**Dataset.**   We investigate our approach based on object recognition and detection tasks. We present a summary of the used dataset in Table 2 and discuss them below.

*Image classification setup*. The image classification setup consists of a traditional image classification framework. The experiments are carried out on three popular standard classification datasets: CIFAR 10 [7], CIFAR 100 [7] and Tiny Imagenet [32]. The CIFAR 10 and CIFAR 100 datasets contain 50,000 training images and 10,000 validation images, with a $32 \times 32$ resolution, over 10 and 100 individual classes respectively. The Tiny Imagenet dataset is made up of 100,000 training images and 10,000 validation images with a resolution of $64 \times 64$. This dataset is made up of 200 individual classes. Classes in the datasets are all mutually exclusive.

*Object detection setup*. For object detection, we follow conventional object detection frameworks that leverage knowledge distillation. We use the combination of the PASCAL VOC 2007 [33] and the PASCAL VOC 2012 [34] dataset for training. The PASCAL VOC 2007 dataset consists of 5011 training/validation images and the PASCAL VOC 2012 contains 11530 training/validation images over the same 20 classes in the PASCAL VOC 2007 dataset. For model evalutation we used the PACAL VOC 2007 test set which consists of 4952 images over 20 different class for evaluating model performance.

**Student-teacher models.**   We used three different learning models, e.g., ResNet18 [29], MobileNetV2 [30] and ShuffleNetV2 [31] for the image classification framework. In Table 3, we include details of those architectures and how we use those in our work. We evaluate these models on different loss functions, datasets, and teacher-student combinations. This includes strong teacher strong student, strong teacher weak student, weak teacher strong student, and weak teacher weak student. In this paper, we follow strong/weak definitions from [76]. Strong models are usually larger than weak models containing many trainable parameters. Because of this, strong models are expected to perform better than weak models. We first train a teacher model, and after that, we train a student model leveraging the output of the teacher model. In a real-life application, a teacher model is supposed to be strong (heavier model) having higher accuracy, and our goal is to train a lighter student model that can perform similarly to the

**Table 2. Summary of datasets used in the study.**

| Dataset | Training Images | Test Images | Classes | Pixel Size | Train Images Per Class | Test images Per Class |
|---|---|---|---|---|---|---|
| CIFAR 10 [7] | 50000 | 10000 | 10 | 32x32 | 5000 | 1000 |
| CIFAR 100 [7] | 50000 | 10000 | 100 | 32x32 | 500 | 100 |
| Tiny ImageNet [32] | 100000 | 10000 | 200 | 64x64 | 500 | 50 |
| PASCAL VOC [33, 34] | 16541 | 4952 | 20 | - | - | - |

**Table 3. Information about different architecture models used in this study.**

| Model | Number of Parameters (M) | Used as Teacher | Used as Student | Used for classification | Used for DFKD | Used for detection |
|---|---|---|---|---|---|---|
| ResNet18 | 11.2 | ✓ | ✓ | ✓ | ✗ | ✗ |
| ShuffleNetV2 | 1.5 | ✓ | ✓ | ✓ | ✗ | ✗ |
| MobileNetV2 | 3.4 | ✗ | ✓ | ✓ | ✗ | ✗ |
| Wrn-40-2 | 2.2 | ✓ | ✗ | ✗ | ✓ | ✗ |
| Wrn-16-1 | 0.2 | ✗ | ✓ | ✗ | ✓ | ✗ |
| VGG16 SSD | 26.2 | ✓ | ✗ | ✗ | ✗ | ✓ |
| MobileNetV2 SSD lite | 3.4 | ✗ | ✓ | ✗ | ✗ | ✓ |

strong teacher model. Therefore, the computational cost/time of inference will reduce in the student model compared to the teacher model.

**Data-free knowledge distillation (DFKD).** For DFKD, we investigate two prominent DFKD techniques: Contrastive Model Inversion (CMI) [77] and Deep Inversion (DI) [57]. For both CMI and DI, we used Wrn-40-2 [78] as the teacher model to train a Wrn-16-1 [78] student model on synthetic CIFAR 10 and CIFAR 100 data.

**Object detection setup.** Object detection looks into classifying multiple classes in a particular image and localizing the class area using a bounding box. Recent development in object detection has brought about new state-of-the-art techniques, Single-Shot Multibox Detector (SSD) [3]. As a result, this study looks into using the stochastic technique on SSD using knowledge distillation. For experiments on object detection, we used VGG16 [3, 79] SSD teacher on a MobileNetV2 SSD lite student [3, 30]. The paper looks into the PASCAL VOC 2007 and PASCAL VOC 2012 training subsets as the dataset for training, and we use PASCAL VOC 2007 as the testing set.

**Evaluation metric.** For classification and detection problems, we evaluate all methods with accuracy and mean average precision (mAP) in percentage, respectively. ***Accuracy***: We employ testing accuracy to gauge the performance of various datasets as an evaluation criterion for classification and data-free knowledge distillation. The percentage (%) of accurately predicted instances in a testing/evaluation set over the total number of instances in the set is used to determine test accuracy. We also report other performance metrics such as Precision, Recall, Kappa, Specificity, Prevalence, and F1 Score to determine the overall performance.

**mAP.** For detection issues that need classification and localization, mAP is the most well-known evaluation technique. The IoU (Intersection Over Union) metric, which measures the intersection of the labeled localization area and the predicted localization area over the union of the labeled localization area and the expected localization area, is used to assess a model created for a detection problem. We consider the prediction accuracy when the IoU is greater than or equal to a predefined threshold. Then, we can determine the precision (the proportion of positives successfully predicted among all positive predictions) and recall (the ratio of positives correctly predicted among all positives labeled) for every class. This calculation uses the calculated IoU for every instance of that class from the area of the predicted and group truth bounding box. The precision and recall values are then plotted as a function of the IoU threshold. Lower IoU values predict higher positive predictions. However, not all positive predictions are correct. As a result, recall increases, and precision decreases. So the curve gets a downward slope. Integrating the curve returns the area under the precision-recall curve, referred to as AP (Average Precision). The average of the AP across all the classes is called mAP, which is used to evaluate detection models.

**Implementation details.** For all experiments, we start with a learning rate of 0.1 and use min-batch gradient descent for parameter training. We run knowledge distillation for

classification, DFKD, and object detection experiments for 100, 200, and 70 epochs, respectively. The learning rate is divided by 0.2 to decay at the $40^{th}$, $60^{th}$, and $80^{th}$ epoch for the classification setup. The learning rate was similarly decayed by 0.2 at $120^{th}$, $160^{th}$, and $180^{th}$ epoch for the data-free knowledge distillation setup. We run the models on an Adam optimizer. The data go through some basic augmentation, such as a random horizontal flip, random rotation of 15 degrees, and a random crop of the image size after 4-pixel padding. We implement the models according to their related papers [3, 29–31, 78, 79]. Also, we train teacher models for 100 epochs and then use them during the knowledge distillation process to learn the student model. We train the teacher models using the cross-entropy loss for both the classification, detection, and DKFD frameworks. We used datasets CIFAR 10, CIFAR 100, Tiny ImageNet, and PASCAL VOC to train the teacher models. For training the student model, we use three loss functions to show the extensiveness of the stochastic technique: distillation loss, L2 loss, and DML loss. To train the student models, we use the output of the teacher model on the loss function to teach the student model. For all experiments, we use an $\alpha$ value of 0.95 and a tuned temperature or the stochastic technique when appropriate. We ran experiments on DKFD and object detection knowledge distillation on official implementation [51, 77]. We have used PyTorch (v1.13 and CUDA 11.3) deep learning framework. We have experimented using a workstation with a single NVIDIA RTX 3070 Ti GPU and 32GB of RAM.

## Main results

**Compared methods.** This paper explores three of the most widely used knowledge distillation methods: Hinton's knowledge distillation (HKD) [26], Caruana's model compression (CMC) [44] and the Deep Mutual Learning Loss (DML) [80]. The CMC and DML loss are enhanced with a temperature hyper-parameter such that a tuned temperature produces better performance than the original loss function. The CMC loss calculates the squared difference between the output and the target logits. The CMC loss does not require additional hyperparameters for softening or weight distribution. However, it does work better with a temperature hyperparameter and is prone to outliers. The DML loss calculates the probabilistic differences between the target and output. The DML loss also does not require softening and weight distribution parameters like the CMC loss. However, it does work better with a temperature hyper-parameter. Here, we summarise all loss equations together.

- CMC loss: $\mathcal{L}_{CMC}(\boldsymbol{h_s}, \boldsymbol{h_m}) = \frac{1}{N} \sum\limits_{c=1}^{n} (\boldsymbol{h_s}(c) - \boldsymbol{h_m}(c))^2$

- Altered CMC loss: $\mathcal{L}_{CMC}(\boldsymbol{h_s}, \boldsymbol{h_m}) = \frac{1}{N} \sum\limits_{c=1}^{n} \left(\frac{\boldsymbol{h_s}(c)}{\tau} - \frac{\boldsymbol{h_m}(c)}{\tau}\right)^2$

- DML loss: $\mathcal{L}_{DML}(\boldsymbol{h_s}, \boldsymbol{h_m}) = \sum\limits_{c=1}^{i} \boldsymbol{h_m}(c) \log \frac{\boldsymbol{h_s}(c)}{\boldsymbol{h_m}(c)}$

- Altered DML loss: $\mathcal{L}_{DML}(\boldsymbol{o_s}, \boldsymbol{o_m}) = \sum\limits_{c=1}^{i} \boldsymbol{o_m}(c) \log \frac{\boldsymbol{o_s}(c)}{\boldsymbol{o_m}(c)}$

**Classification.** Table 4 demonstrates results obtained for image classification standard classification datasets (CIFAR 10, CIFAR 100, and Tiny ImageNet) using a tuned temperature and the stochastic technique. *(1)* Performance obtained for the stochastic technique is similar to a tuned temperature for the HKD loss and DML loss. The CMC loss performs better using a tuned temperature than the stochastic temperature. This is because the CMC loss does not use softening to transfer knowledge. Nonetheless, the power consumption and $CO_2$ production

**Table 4. Carbon footprints and performance for different KD approaches.** Here, for image recognition, we use ResNet18 (teacher) and MobileNetV2 (student) models. For object detection, we use VGG16 (teacher) and MobileNetV2 (student). We report average results after running the same program five times. ↑ (↓) means higher (lower) is better.

| Loss | Method | CIFAR 10 | | | | CIFAR 10 | | | |
|---|---|---|---|---|---|---|---|---|---|
| | | Accuracy (%) ↑ | GFLOPs (M) ↓ | Energy (kWh) ↓ | $CO_2$_eq (g) ↓ | Accuracy (%) ↑ | GFLOPs (M) ↓ | Energy (kWh) ↓ | $CO_2$_eq (g) ↓ |
| HKD | [26] | **91.78 ± 0.14** | 128.30 | 3.99 | 1173.44 | **69.70 ± 0.12** | 128.40 | 3.80 | 1170.40 |
| | Ours | 91.67 ± 0.23 | **6.42** | **0.21** | **61.76** | 69.40 ± 0.37 | **6.42** | **0.20** | **61.60** |
| CMC | [44] | **92.35 ± 0.24** | 128.30 | 3.93 | 1192.81 | **73.56 ± 0.17** | 128.40 | 3.92 | 1164.7 |
| | Ours | 91.65 ± 0.13 | **6.42** | **0.21** | **61.17** | 72.23 ± 0.24 | **6.42** | **0.21** | **61.30** |
| DML | [80] | 91.00 ± 0.04 | 128.30 | 4.01 | 1161.00 | **72.90 ± 0.04** | 128.40 | 3.74 | 1109.40 |
| | Ours | **91.05 ± 0.13** | **6.42** | **0.20** | **58.05** | 72.56 ± 0.15 | **6.42** | **0.19** | **58.04** |

| Loss | Method | Tiny ImageNet | | | | PASCAL VOC | | | |
|---|---|---|---|---|---|---|---|---|---|
| | | Accuracy (%) ↑ | GFLOPs (M) ↓ | Energy (kWh) ↓ | $CO_2$_eq (g) ↓ | mAP (%) ↑ | GFLOPs (M) ↓ | Energy (kWh) ↓ | $CO_2$_eq (g) ↓ |
| HKD | [26] | 60.46 ± 0.07 | 975 | 19.09 | 5595.31 | 65.45 ± 0.07 | 1371 | 31.25 | 9235.39 |
| | Ours | **60.53 ± 0.31** | **48.75** | **1.04** | **304.88** | **65.48 ± 0.15** | **72.17** | **1.69** | **499.21** |
| CMC | [44] | **62.76 ± 0.17** | 975 | 20.21 | 5949.93 | **64.87 ± 0.13** | 1371 | 32.27 | 9527.55 |
| | Ours | 59.84 ± 0.11 | **48.75** | **1.00** | **293.10** | 64.67 ± 0.12 | **72.17** | **1.70** | **501.45** |
| DML | [80] | 63.30 ± 0.11 | 975 | 19.38 | 5748.64 | **65.10 ± 0.09** | 1371 | 30.24 | 8903.52 |
| | Ours | **63.35 ± 0.07** | **48.75** | **1.02** | **302.56** | 64.98 ± 0.24 | **72.17** | **1.68** | **494.64** |

when using the stochastic technique are drastically low compared to tuning for knowledge distillation. *(2)* The FLOPs count on every dataset is similar for its respective losses as the models used are identical for all the losses. The losses do not add to the FLOPs count of the model. The FLOPs of the models increase for larger datasets due to pixel size and instances. The results of the CIFAR 100 dataset are similar in energy consumption and carbon footprint production compared to the CIFAR 10 dataset, as both datasets have the same number of instances in their train and validation set and the same pixel size. *(3)* The performance measure for every dataset is a test side performance of the evaluation set of the dataset, which consists of unseen images. The efficiency metrics (FLOPs, Energy, $CO_2$_eq) are measured during training to estimate the training efficiency of tuning instead of using the stochastic technique. *(4)* The stochastic technique outperforms tuning heavily in terms of efficiency and produces similar results in comparison. Using the CMC loss on the CIFAR 10 dataset, the tuned temperature performs significantly better, achieving an accuracy of 92.35% compared to the stochastic technique at 91.65%. However, tuning the framework for the CMC loss consumes about 20 times more energy and produces $CO_2$ equal to a car running for 9.91km. On the CIFAR 10 dataset, the stochastic technique performs similarly to a tuned temperature. However, the stochastic approach consumes 0.20kWh of energy and produces a negligible carbon footprint of 58.05g, whereas tuning requires 4.0kWh of energy. The same trend can be seen across all the datasets. HKD loss on the Tiny Imagenet data performs almost equally for both the tuned and stochastic temperature, with a difference of 0.06% favoring the stochastic technique. Additionally, the stochastic approach consumes 1.04kWh of energy and produces 304.88g of $CO_2$, which is negligible compared to tuning, which consumes 19.09kWh of energy and produces 5595.31g of $CO_2$. The CMC loss for Tiny ImageNet performs very well compared to the stochastic technique. For the DML loss on the Tiny ImageNet dataset, the stochastic approach performs better by 0.05% against a tuned temperature. Moreover, tuning produces 19.38g of $CO_2$ and consumes 1.02kWh of energy which is significantly lower than tuning, which requires 19.38kWh to be tuned and makes 5748.64g of $CO_2$ that is comparable to driving a car for 47.8km. *(5)* Performance of the stochastic technique and tuned temperature are pretty

**Table 5. Different performance metrics of ResNet18-MobileNetV2 as teacher-student model combination using CIFAR 10 dataset.** In addition to maintaining low carbon footprints, `Ours` method performs similarly to the KD [26] method.

| Model | Accuracy | Precision | Recall | Specificity | Kappa | Prevalence | F1 |
|---|---|---|---|---|---|---|---|
| Teacher | 94.80 | 0.95 | 0.95 | 0.95 | 0.94 | 0.09 | 0.95 |
| KD [26] | 91.67 | 0.92 | 0.92 | 0.92 | 0.90 | 0.09 | 0.92 |
| Ours | 91.78 | 0.92 | 0.92 | 0.92 | 0.91 | 0.09 | 0.92 |

similar across the HKD loss as the HKD loss primarily uses temperature for knowledge transfer. Quite a similar trend can also be seen for DML loss on all the datasets, as the DML loss can also be used with a temperature for knowledge transfer. For the CMC loss, performance is higher with a tuned temperature. This is because the CMC loss is generally not used with a temperature hyperparameter. It is worth noting that the CMC loss performs better with a tuned temperature than with its base form. The DML loss serves better with a tuned temperature and the stochastic technique against the base DML loss. (6) The stochastic technique contributes towards Green AI by replacing the requirement of tuning in model compression frameworks. The efficiency of the stochastic technique is considered as opposed to tuning and a commendable alternative to use.

In Table 5, we further report Precision, Recall, Kappa, Specificity, Prevalence, and F1 Score for ResNet18-MobileNetV2 teacher-student combination on the CIFAR 10 dataset. The teacher model has an accuracy of 94.8% and produces average precision and recall of 0.95, exhibiting that the models have learned all the classes properly. Similarly, the student models achieve nearly equal recall and precision (0.92), corresponding to the accuracy of 91.67% (Tuned) and 91.78% (Ours). We also compute the F1 score which is the harmonic mean of precision and recall. Here, F1 scores of the Teacher model (0.95), KD model (0.92), and Our model (0.92) effectively show that the training is not biased. The Kappa score of our model (0.91) also establishes that the performances and outputs produced are highly agreeable with the labels of the dataset.

In Fig 4, we further visualize the output of No compression, KD [26], and `Ours` methods. The output logit shown in Fig 4(a) indicates that all methods have correctly classified a sample bird image input because the class bird gets the highest score. Fig 4(b) showcases Grad-Cam images from the last layer of the MobileNetV2 architecture. `Ours` and KD [26] methods produced similar Grad-cam visualization, suggesting that both have learned from the same teacher. High values at the diagonal positions of the confusion matrices in Fig 4(c) demonstrate that `Ours` method is equally successful with No compression and KD [26] methods.

**Detection.** The PASCAL VOC dataset results in Table 4 (second row) show performances for object detection. For the HKD loss, our proposed stochastic technique is similar to a tuned temperature with a minor accuracy difference of 0.03%. However, traditional tuning uses 31.25 kWh of energy. Tuning such an object detection framework produces 9235.39g of $CO_2$, equivalent to a car running for 78.66 Km. Comparatively, our stochastic technique minimizes the $CO_2$ consumption by about 18 times and requires only 1.69 kWh of energy. For the CMC loss, the tuned temperature performs better (64.87% accuracy) compared to the 64.67% accuracy of the stochastic technique. However, the cost of tuning is similarly high, requiring 1.70 kWh of energy and producing 9527.55g of $CO_2$ as opposed to the stochastic approach. Our method requires a reduced consumption of energy and $CO_2$ production of 1.70kwh and 9527.55g, respectively. Similarly, the DML loss performs quite similarly in terms of performance compared to the stochastic technique. Still, the traditional KD approach is costly in energy and carbon footprint compared to the stochastic approach.

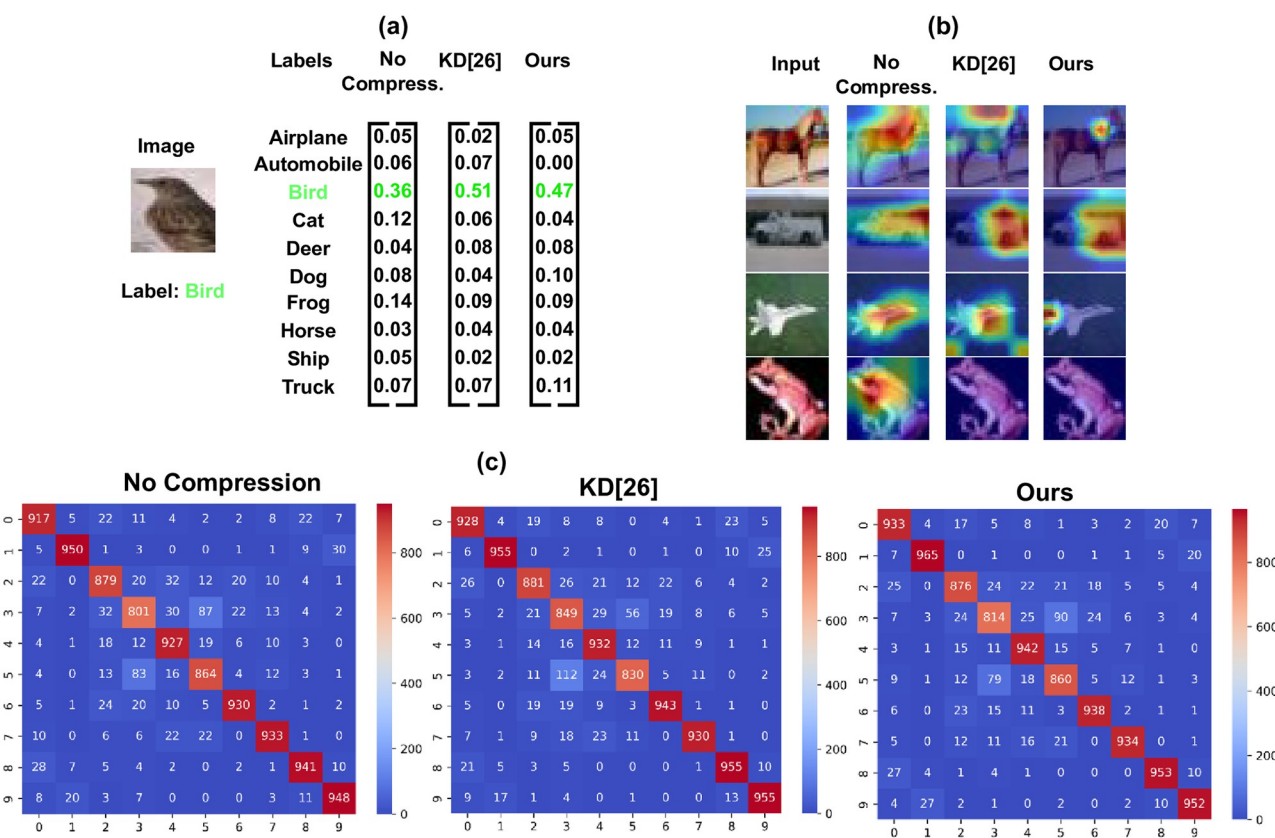

**Fig 4. Visual illustration of output produced by No compression, KD [26] and Ours methods.** (a) Output logits for a sample 'bird' image from the CIFAR 10. (b) Grad-Cam visualization of four sample images from the CIFAR 10 using the MobileNetV2. ResNet18 is used as. (c) Confusion matrices of prediction.

## Data-free knowledge distillation (DFKD) results

Knowledge distillation can be a useful model compression technique. However, it also brings about data requirement constraints, which might be unavailable due to privacy concerns [56, 58, 81]. As a result, DFKD is widely popular since it removes training data constraints by generating synthetic/artificial training images/instances [82] for many computer vision [83] and natural language processing [84] tasks. Due to the significance of DFKD, this paper also looks into the implementation and performance of DFKD frameworks. For experiments on DFKD, we use two prominent DFKD techniques: Contrastive Model Inversion (CMI) [77] and Deep Inversion (DI) [57]. The CMI looks into inverting a model to create a diverse distinguishable group of synthetic images using data diversity as an optimization objective. DI also produces artificial images by inverting learning models but optimizing inputs and regularizing feature maps using information stored in batch normalization. Table 6 represents experimental results obtained on the CMI and DI data-generation techniques on CIFAR 10 and CIFAR 100 datasets using a tuned temperature against our proposed stochastic approach. For DI, using the stochastic temperature on CIFAR 10 dataset had a similar performance to fine-tuning (83.09% compared to 83.04%). Nonetheless, fine-tuning produces a massive carbon footprint of 1369.46g of $CO_2$ compared to 72.07g of $CO_2$ produced by the stochastic technique. The stochastic process performs similarly to the tuned temperature, with a difference of 0.25% for CIFAR 10 and 0.31% for CIFAR 100. The stochastic approach produces results higher than the

**Table 6. Accuracy and carbon footprints obtained on different data-free knowledge distillation (DFKD) techniques.** ↑ (↓) means higher (lower) is better.

| DFKD | Method | CIFAR 10 | | | | CIFAR 10 | | | |
|---|---|---|---|---|---|---|---|---|---|
| | | Accuracy (%) ↑ | GFLOPs (M) ↓ | Energy (kWh) ↓ | $CO_2$_eq (g) ↓ | Accuracy (%) ↑ | GFLOPs (M) ↓ | Energy (kWh) ↓ | $CO_2$_eq (g) ↓ |
| CMI | Tuned | **87.38** | 149.18 | 4.56 | 1353.66 | **57.91** | 149.18 | 4.94 | 1390.8 |
| | Ours | 87.13 ± 0.21 | **7.85** | **0.24** | **71.24** | 57.60 ± 0.31 | **7.85** | **0.26** | **73.20** |
| DI | Tuned | 83.04 | 149.18 | 4.75 | 1369.46 | **61.34** | 149.18 | 4.75 | 1393.08 |
| | Ours | **83.09 ± 0.14** | **7.85** | **0.25** | **72.07** | 61.24 ± 0.21 | **7.85** | **0.25** | **72.32** |

second most optimum temperature yet consumes 19 times less power and produces likewise less $CO_2$ than the tuned technique. The stochastic design optimizes hyperparameters for data-free knowledge distillation without a costly tuning process and delivers performance similar to the tuning method with lower cost, time, and carbon production.

**Varying student-teacher models.** To prove the robustness of our proposed method, in Table 7, we experiment with different teacher-student combinations using the Tiny ImageNet dataset. Our proposed stochastic method can work satisfactorily irrespective of strong (large models) or weak (light models) students and teachers. When using a strong teacher and a weak student for model compression, the experiment using a stochastic temperature has a 0.07% increase in accuracy compared to a tuned approach. When using self-knowledge distillation using two light models, the tuned experiments have a test accuracy of 60.32%, and the stochastic process has a test accuracy of 60.38%. Even such light model combinations can produce the same amount of $CO_2$ as an average car if it travels for 37.3 Km. Table 7 also showcases that with an increase in the number of model parameters for the student or teacher, there is an increase in performance. The change in FLOPs on the same dataset is due to using different student-teacher architectures. The stochastic technique produces satisfactory results and outweighs tuning in efficiency significantly.

## Model compression using quantization

In addition to KD, model quantization can be another way of model compression. In Table 8, we compare a quantization method named quantization-aware-training [85] with the KD methods used in this paper. Deep learning models store parameters as floating points (32 bits or 64 bits) to achieve high precision and accuracy. However, quantization-based methods quantize the precision of input and parameters by reducing bit-width to integers (8-bit), reducing the model size, inference time, and performance. Table 8 showcases that the KD technique consumes 3.99kWh of energy and produces 1173.44g of $CO_2$, and performs with an

**Table 7. Performance of different student-teacher combinations using Tiny ImageNet dataset.** Our stochastic method consistently performs a lower carbon footprint than the tuned KD approach, keeping similar accuracy. ↑ (↓) means higher (lower) is better.

| | Weak Teacher: ShuffleNetV2 (1.5M parameters) | | | | Strong Teacher: ResNet18 (11.2M parameters) | | | |
|---|---|---|---|---|---|---|---|---|
| | Weak Student: ShuffleNetV2 (1.5M parameters) | | | | Weak Student: ShuffleNetV2 (1.5M parameters) | | | |
| Method | Accuracy (%) ↑ | GFLOPs (M) ↓ | Energy (kWh) ↓ | $CO_2$_eq (g) ↓ | Accuracy (%) ↑ | GFLOPs (M) ↓ | Energy (kWh) ↓ | $CO_2$_eq (g) ↓ |
| Tuned | 60.32 ± 0.08 | 155.48 | 15.26 | 4489.89 | 60.46 ± 0.07 | 975.23 | 19.09 | 5595.31 |
| Ours | **60.38 ± 0.11** | **7.77** | **0.81** | **238.41** | **60.53 ± 0.31** | **48.75** | **1.04** | **304.88** |
| | Weak Student: ShuffleNetV2 (1.5M parameters) | | | | Strong Student: ResNet18 (11.2M parameters) | | | |
| | Strong Student: ResNet18 (11.2M parameters) | | | | Strong Student: ResNet18 (11.2M parameters) | | | |
| Method | Accuracy (%) ↑ | GFLOPs (M) ↓ | Energy (kWh) ↓ | $CO_2$_eq (g) ↓ | Accuracy (%) ↑ | GFLOPs (M) ↓ | Energy (kWh) ↓ | $CO_2$_eq (g) ↓ |
| Tuned | **63.66 ± 0.12** | 1057 | 32.30 | 9503.34 | **63.69 ± 0.06** | 1877.63 | 34.47 | 10143.34 |
| Ours | 63.38 ± 0.23 | **52.85** | **1.58** | **466.17** | 63.45 ± 0.17 | **93.83** | **1.85** | **544.94** |

**Table 8. Comparison among quantization and KD methods for model compression.** Experiments are done on MobileNetV2 architecture on the CIFAR 10 dataset. KD techniques use ResNet18 as the teacher. `Ours` method achieves the best performance in both accuracy and carbon footprint metrics.

| Method | Accuracy (%) ↑ | GFLOPs (M) ↓ | Energy (kWh) ↓ | $CO_2$_eq (g) ↓ |
|---|---|---|---|---|
| No compression/KD | 90.52 | 2.16 | 0.16 | 45.02 |
| Quantization [85] | 89.97 | **2.46** | 0.40 | 124.42 |
| KD [26] | 91.62 | 128.30 | 3.99 | 1173.44 |
| `Ours` | **91.78** | 6.42 | **0.21** | **61.76** |

accuracy of 91.62%. The MobileNetV2 model produced by quantization-aware training builds a smaller and faster model but lacks performance with an accuracy of 89.97%. The quantized model consumes significantly lower energy as opposed to KD. Nonetheless, our proposed stochastic technique consumes lower power (0.21kWh), produces a smaller carbon footprint overall, and performs similarly to both methods.

## Conclusion

The deployment of large models is infeasible to mobile and edge computing devices. Knowledge distillation provides a solution to this issue by generating a significantly lighter and deployable model on mobile and edge computing devices. However, we demonstrate that this suitability often comes at the expense of substantial environmental costs, largely ignored in deep learning literature. The objective of this article is threefold: (1) to investigate environmental costs for deep learning model compression using knowledge distillation, (2) to propose a stochastic approach as a means to mitigate carbon footprints for the knowledge distillation process, and (3) to conduct extensive experiments that demonstrate the suitability of the proposed stochastic approach. We propose that deep learning research should not be measured only by performance metrics such as accuracy and mAP, rather the performance metrics should also include environmental costs. Extensive experiments were conducted using various student-teacher model combinations (based on ResNet18, MobileNetV2, and ShuffleNetV2) to solve image classification and object detection problems. We estimated the carbon footprints of the overall computation process for each student-teacher model combination in terms of FLOPs count in millions, energy consumption in kWh, and CO2 equivalent in grams. Based on our empirical findings, KD consumes 13 to 18 times more carbon than the heavier teacher model. The repetitive tuning of hyperparameters is primarily responsible for such astronomical environmental costs. This article investigates the environmental impact of model compression techniques based on knowledge distillation and proposes a stochastic approach that requires less computation without sacrificing performance. Empirical results demonstrate that the proposed stochastic approach applied on CIFAR 100 dataset consumes 0.20kWh of energy and emits 61.60g of $CO_2$ while the tuning approach consumes 3.80kWh of energy and emits 1170.40g of CO2 with a performance difference favoring the tuning approach of 0.30%. Similarly, for the Tiny ImageNet dataset, the performance accuracy of the stochastic approach is 60.53 percent, compared to tuning's performance accuracy of 60.46 percent. However, the stochastic technique only consumes 1.04kWh of energy and emits 304.88g of CO2, whereas tuning approach emits a massive amount of carbon footprint (5,595.31g) and uses a tremendous amount of energy (19.09kWh). In future, we intend to investigate the carbon footprint of additional deep-learning tasks, such as continuous learning, domain adaptation, and meta-learning, in which the same model undergoes multiple training rounds.

## Author Contributions

**Conceptualization:** Kazi Rafat, Shafin Rahman, Nabeel Mohammed.

**Data curation:** Kazi Rafat.

**Formal analysis:** Kazi Rafat.

**Funding acquisition:** Fuad Rahman.

**Investigation:** Kazi Rafat.

**Methodology:** Kazi Rafat, Sadia Islam, Abdullah Al Mahfug, Md. Ismail Hossain, Nabeel Mohammed.

**Project administration:** Fuad Rahman, Sifat Momen, Nabeel Mohammed.

**Software:** Kazi Rafat.

**Supervision:** Shafin Rahman, Nabeel Mohammed.

**Validation:** Kazi Rafat.

**Visualization:** Kazi Rafat, Shafin Rahman, Nabeel Mohammed.

**Writing – original draft:** Kazi Rafat, Sadia Islam, Abdullah Al Mahfug, Shafin Rahman, Nabeel Mohammed.

**Writing – review & editing:** Kazi Rafat, Sifat Momen, Shafin Rahman, Nabeel Mohammed.

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
