## [Decision Letter · Decision Letter 0]

13 Feb 2023

PONE-D-23-01498Mitigating Carbon Footprint for Knowledge Distillation Based Deep Learning Model CompressionPLOS ONE

Dear Dr. Rahman,

Thank you for submitting your manuscript to PLOS ONE. After careful consideration, we feel that it has merit but does not fully meet PLOS ONE’s publication criteria as it currently stands. Therefore, we invite you to submit a revised version of the manuscript that addresses the points raised during the review process.

We look forward to receiving your revised manuscript.

Kind regards,

Süleyman Eken, Ph.D.

Academic Editor

PLOS ONE

Journal Requirements:

3. We note that Figure 2 in your submission contain copyrighted images. All PLOS content is published under the Creative Commons Attribution License (CC BY 4.0), which means that the manuscript, images, and Supporting Information files will be freely available online, and any third party is permitted to access, download, copy, distribute, and use these materials in any way, even commercially, with proper attribution. For more information, see our copyright guidelines: http://journals.plos.org/plosone/s/licenses-and-copyright.

Reviewers' comments:

Reviewer's Responses to Questions

**Comments to the Author**

1. Is the manuscript technically sound, and do the data support the conclusions?

Reviewer #1: Yes

Reviewer #2: Partly

2. Has the statistical analysis been performed appropriately and rigorously? 

Reviewer #1: Yes

Reviewer #2: Yes

3. Have the authors made all data underlying the findings in their manuscript fully available?

Reviewer #1: No

Reviewer #2: Yes

4. Is the manuscript presented in an intelligible fashion and written in standard English?

Reviewer #1: Yes

Reviewer #2: Yes

5. Review Comments to the Author

Reviewer #1: Knowledge distillation based methods are very important nowadays due to providing an efficent computation for small hardware resources. The paper is written in this direction. In general, I like the paper including theoretical background and experimental results. Results are clear to prove their hypothesis. On the other hand, it would be better if the following revisions are applied in the next version of the manuscript.

1- There is another way to compress big DL model called model quantization/pruning. As is known, each parameter is stored in a 32-bit data structure. You can simply quantize/prune the model using the state-of-the-art techniques and compare them with knowledge distillation. It would increase the popularity of the paper.

2- There are no sample images so it is very hard to interpret results. You can extend the paper with the results and compare in terms of human vision.

Reviewer #2: The authors examined the environmental costs (carbon footprints) for deep learning model compression using knowledge distillation. The authors have extensively experimented with different combinations of student-teacher models based on the architectures of ResNet18, MobileNetV2 and ShuffleNetV2, as well as CIFAR10, CIFAR 100, Tiny Imagenet and PASCAL VOC reported both object recognition and detection problems using their datasets. Researchers have taken up a very interesting study indeed. My reviews and suggestions about their publications are listed;

More numeric values should be given in the abstract. The abstract should include the context or background information for your research; the general topic under study; the specific topic of your research; why is it important to address these questions; the significance or implications of your findings or arguments. It must also contain more numeric values. Please highlight your contribution. Reorganize the abstract to conclude: (a) The overall purpose of the study and the research problems you investigated. (b) The basic design of the study. (c) Major findings or trends found as a result of the study. (d) A brief summary of your interpretations and conclusions.

Some table and figure texts are really long. These need to be shortened. Figure texts and main texts should be separated from each other.

The mathematical background for the proposed method presented in the "Stochastic solution" section should be presented.

Although some evaluation criteria are given in the article, It should be well supported by Precision, Recall (sensitivity), Specificity, Prevalence, Kappa, and F1-score. These results need to be analyzed, tabulated, presented graphically, and interpreted.

rewrite the conclusion with following comment: (a) Highlight your analysis and reflect only the important points for the whole paper. (b) Mention the implication in the last of this section. Please, carefully review the manuscript to resolve these issues. (c) This section should be supported with numerical values.

You should use a more academic language. The authors should polish the manuscript to improve its writing. The quality of the article should be increased. English must be strongly revised to make the paper even readable.

6. PLOS authors have the option to publish the peer review history of their article (what does this mean?). If published, this will include your full peer review and any attached files.

Reviewer #1: No

Reviewer #2: No

---

## [Author Response · Author response to Decision Letter 0]

30 Mar 2023

We thank the editor and reviewers for considering our paper. 

Our detailed response is uploaded in the Attach file section, addressing every editorial and review comments.

---

## [Decision Letter · Decision Letter 1]

28 Apr 2023

Mitigating Carbon Footprint for Knowledge Distillation Based Deep Learning Model Compression

PONE-D-23-01498R1

Dear Dr. Rahman,

We’re pleased to inform you that your manuscript has been judged scientifically suitable for publication and will be formally accepted for publication once it meets all outstanding technical requirements.

Kind regards,

Süleyman Eken, Ph.D.

Academic Editor

PLOS ONE

Additional Editor Comments (optional):

Reviewers' comments:

Reviewer's Responses to Questions

**Comments to the Author**

1. If the authors have adequately addressed your comments raised in a previous round of review and you feel that this manuscript is now acceptable for publication, you may indicate that here to bypass the “Comments to the Author” section, enter your conflict of interest statement in the “Confidential to Editor” section, and submit your "Accept" recommendation.

Reviewer #1: All comments have been addressed

Reviewer #2: All comments have been addressed

2. Is the manuscript technically sound, and do the data support the conclusions?

Reviewer #1: Yes

Reviewer #2: Yes

3. Has the statistical analysis been performed appropriately and rigorously? 

Reviewer #1: Yes

Reviewer #2: Yes

4. Have the authors made all data underlying the findings in their manuscript fully available?

Reviewer #1: Yes

Reviewer #2: Yes

5. Is the manuscript presented in an intelligible fashion and written in standard English?

Reviewer #1: Yes

Reviewer #2: Yes

6. Review Comments to the Author

Reviewer #1: I would like to thank the authors for their efforts on the last version. I think the manuscript is ready to meet with the readers.

Reviewer #2: The authors have progressed in improving the paper compared to previous versions of the paper (PONE-D-23-01498 & PONE-D-23-01498_R1). When the previous and revised versions of the paper are evaluated together, it is seen that the authors make the corrections requested by the referees and show the necessary sensitivity in the revision of the paper in line with the comments. In the revised version of the paper, almost all the comments have been considered and addressed by the authors.

The response to reviewers file is well-prepared. The changes made by the authors in line with the opinions/suggestions/evaluations of the referees can be tracked. The existing organization and spelling problems in the previous version of the article have been fixed. In the revised version, the clarity and follow-up of the study have been increased. In addition, the article has been carefully reviewed for grammatical and typos.

As a result, my concerns on the previous version of the paper have disappeared with the explanations made by the authors, as well as the revision they have made.

This revision is sufficient, and it is possible to evaluate the paper for publication after preparation according to PLOS ONE template.

7. PLOS authors have the option to publish the peer review history of their article (what does this mean?). If published, this will include your full peer review and any attached files.

Reviewer #1: No

Reviewer #2: No

---

## [Editor Report · Acceptance letter]

5 May 2023

PONE-D-23-01498R1 

Mitigating Carbon Footprint for Knowledge Distillation Based Deep Learning Model Compression 

Dear Dr. Rahman:

I'm pleased to inform you that your manuscript has been deemed suitable for publication in PLOS ONE. Congratulations! Your manuscript is now with our production department. 

Kind regards, 

on behalf of

Assoc. Prof. Dr. Süleyman Eken 

Academic Editor

PLOS ONE